# Liquid Biopsy in Early-Stage Lung Cancer: Current and Future Clinical Applications

**DOI:** 10.3390/cancers15102702

**Published:** 2023-05-10

**Authors:** Olivia Vandekerckhove, Kristof Cuppens, Karin Pat, Bert Du Pont, Guy Froyen, Brigitte Maes

**Affiliations:** 1Department Pulmonology and Thoracic Oncology, Jessa Hospital, 3500 Hasselt, Belgium; 2Department Thoracic Oncology, The Netherlands Cancer Institute, Amsterdam and Leiden University Medical Center, 2333 Leiden, The Netherlands; 3Faculty of Medicine and Life Sciences—LCRC, Hasselt University, 3590 Diepenbeek, Belgium; 4Department Thoracic and Vascular Surgery, Jessa Hospital, 3500 Hasselt, Belgium; 5Laboratory for Molecular Diagnostics, Department Laboratory Medicine, Jessa Hospital, 3500 Hasselt, Belgium

**Keywords:** liquid biopsy, circulating tumor DNA, early stage, non-small cell lung cancer, review, molecular diagnostics, targeted therapy, screening, minimal residual disease

## Abstract

**Simple Summary:**

Early-stage disease non-small cell lung cancer has better outcomes than advanced disease, but 5-year survival rates can drop to approximately 50% in cases of increased tumor size, local extension, or nodal spread. The use of liquid biopsies to enhance diagnosis, optimize perioperative systemic treatments, and allow early detection of relapse is a possible strategy to reduce this burden. This review aims to present the current evidence on clinical applications of liquid biopsies in early-stage non-small cell lung cancer and highlight opportunities for future applications.

**Abstract:**

Lung cancer remains the leading cause of cancer death worldwide, with the majority of cases diagnosed in an advanced stage. Early-stage disease non-small cell lung cancer (NSCLC) has a better outcome, nevertheless the 5-year survival rates drop from 60% for stage IIA to 36% for stage IIIA disease. Early detection and optimized perioperative systemic treatment are frontrunner strategies to reduce this burden. The rapid advancements in molecular diagnostics as well as the growing availability of targeted therapies call for the most efficient detection of actionable biomarkers. Liquid biopsies have already proven their added value in the management of advanced NSCLC but can also optimize patient care in early-stage NSCLC. In addition to having known diagnostic benefits of speed, accessibility, and enhanced biomarker detection compared to tissue biopsy, liquid biopsy could be implemented for screening, diagnostic, and prognostic purposes. Furthermore, liquid biopsy can optimize therapeutic management by overcoming the issue of tumor heterogeneity, monitoring tumor burden, and detecting minimal residual disease (MRD), i.e., the presence of tumor-specific ctDNA, post-operatively. The latter is strongly prognostic and is likely to become a guidance in the postsurgical management. In this review, we present the current evidence on the clinical utility of liquid biopsy in early-stage lung cancer, discuss a selection of key trials, and suggest future applications.

## 1. Introduction

Lung cancer is the deadliest and second-most common cancer worldwide, representing 12% of new cases and 21% of cancer-related deaths. Overall survival at 5 years from initial diagnosis drops from 60% for stage IIA to 36% for stage IIIA and further down to 6% for stage IV (metastatic disease) lung cancer [1]. Early diagnosis, optimized perioperative systemic treatments, and timely detection of relapse are strategies to improve lung cancer survival.

Diagnosis and treatment of early-stage NSCLC are streamlined based on patient profile and preference, TNM staging, and histopathological and molecular analysis. The latter is commonly performed on a tumor tissue biopsy or a surgical specimen, which requires invasive diagnostic methods, often at the cost of clinically valuable time. Thanks to advances in molecular diagnostics and the growing availability of targeted therapies, the application of liquid biopsy (LB) could not only facilitate the diagnostic process but could also identify patients who might benefit from directed systemic therapy.

LB is the real-time analysis of malignant tumor cells or tumor cell products released into the blood or other body fluids by primary or metastatic tumor lesions [2,3,4]. Circulating tumor DNA (ctDNA), cell-free tumor DNA (cfDNA), circulating tumor cells (CTC), micro-RNAs (miRNAs), and DNA methylation signatures are the most commonly detected tumor-related liquid biomarkers [5]. ctDNA analysis is currently the most widely implemented in clinical care. cfDNA can—unlike ctDNA—also be found in healthy subjects, although levels are higher in cancer patients. Both ctDNA and cfDNA are associated with disease progression and are more abundant than CTCs. The presence of CTCs is considered a prognostic biomarker since it can help predict disease progression in cancer patients, for CTCs are one of the mechanisms through which cancer spreads from one organ to another. miRNAs are promising early detection biomarkers since their detection in LB samples was reported to distinguish NSCLC patients from healthy subjects, which could be of great value in lung cancer screening programs. Genetic and epigenetic mutations, such as DNA methylation, are associated with various cancer types, including lung cancer, and could also be used for screening and early detection of NSCLC.

LB has already proven its added value in the management of many cancers [6,7,8,9] including advanced NSCLC [10]. In advanced NSCLC, plasma ctDNA analysis is considered complementary to tissue biopsy and in some cases even the preferred method. Utilizing ctDNA in addition to tissue biopsy increases the detection rate of an actionable mutation in oncogene-driven NSCLC by 48% [11]. Moreover, it is the preferred method to detect mechanisms of acquired resistance to targeted agents (e.g., EGFR tyrosine kinase inhibitors (TKIs)) as LB overcomes the issue of tumor heterogeneity. Finally, in non-oncogene-driven NSCLC, LB can be used to determine tumor mutation burden (TMB) as a biomarker for response and improved clinical outcome when treated with immune checkpoint inhibitors [12,13,14]. The role of blood-based TMB in therapy decisions, however, remains unclear. In addition to these advantages of the use of LB in advanced NSCLC, the implementation of LB in the diagnostic setting of early-stage NSCLC could also encompass some other benefits (Figure 1). It could facilitate the diagnostic process in impossible-to-reach nodules, increase the detection of all pathogenic variants, and shorten the time to definitive treatment. Furthermore, there may be a future role for LB in the screening, prognosis, and therapy of early-stage NSCLC, which will also be discussed here.

The use of LB also faces some limitations. There is a risk of false negative results in cases of gene rearrangements such as fusions or translocations (e.g., EML4-ALK translocation) or when the quantity of ctDNA is limited [15]. For example, estimated clonal variant allele frequencies (VAF) of ctDNA are only 0.008% (95% CI 0.002–0.03%) for cT1b, 0.1% (95% CI 0.06–0.18%) for cT1c, and 1.4% (95% CI 0.62–3.1%) for cT3 NSCLC, whereas the most sensitive current LB technologies typically have a detection limit of 0.01% [16]. With decreasing disease stage, the chance of detecting ctDNA diminishes [17].

There is also a risk of false positive results based on nontumor mutations. Most of these detected mutations reflect clonal hematopoiesis and are non-recurrent. In comparison to tumor-derived mutations, clonal hematopoiesis mutations occur on longer circulating DNA fragments and lack mutational signatures associated with tobacco smoking [17,18]. These “false variants” also appear to be elevated in other diseases, such as acute myocardial infarction and interstitial lung disease [19].

Lastly, the major challenges limiting the use of LB are the availability of clinically valid assays and reimbursement, hindering broad implementation in the clinic.

Keeping in mind these advantages and limitations, LB will become a useful tool to enhance the management of early-stage NSCLC. In this review, we present the current evidence on clinical applications of LB in early-stage lung cancer and suggest future applications (Figure 1).

## 2. Diagnosis of Early-Stage NSCLC

Currently, tissue biopsy still is the golden standard for the diagnosis and molecular evaluation of NSCLC. However, it is not always possible to obtain tumor tissue in a minimal-invasive way. Particularly in the lower stages of early-stage NSCLC, it often proves to be more difficult if not impossible to obtain tissue from small (usually peripheral) nodules without surgical intervention. Furthermore, obtaining sufficient tumor tissue minimally invasively, allowing reliable detection of pathogenic (epi)genetic variations, is even more challenging. The first and probably most important diagnostic advantage of LB in early-stage NSCLC is therefore its non-invasive character, alongside good repeatability [20,21] (Table 1).

The second advantage of LB in comparison to tissue biopsy alone is that it has the potential to speed up confirmation of a presumed lung cancer diagnosis. A prospective study in 282 patients with untreated metastatic NSCLC demonstrated that ctDNA analysis identified oncogenic variants on average 6 days before standard-of-care tissue genotyping, resulting in 9 versus 16 days of turnaround time (TAT) and thereby shortening the time to treatment significantly (median 18 days with liquid biopsy vs. 31 days with tumor tissue molecular profiling, *p* = 0.0008) [11]. These results were in line with a retrospective analysis, showing a median TAT in favor of LB [22,23]. The time between the molecular analysis request on a tissue biopsy and receiving the molecular report on the last biomarker was on average 21 (range 5–66) days. In LB, the median time from the blood draw to the results was 10.5 (range 7–19) days. To shorten the time to definitive treatment, an LB-based diagnosis of early-stage NSCLC could prove to be a useful tool. A wide range in TAT is noted, but institutions with dedicated staff and standard operating procedures facilitating reflex testing can shorten TAT significantly both in tissue and liquid testing.

Complementary to the above-mentioned advantages, using LB enables tissue preservation for evaluation of other biomarkers that currently cannot be detected through liquid biopsy, such as PD-L1 but also gene fusions or rearrangements [11]. This complementary strategy could facilitate the assessment of all guideline-recommended therapeutic and prognostic biomarkers.

The fourth advantage could be an increased detection of pathogenic variants. In advanced NSCLC, LB has already been demonstrated to enhance the detection of oncogenic alterations, as mentioned above [7,8]. With improving detection sensitivity and a broadening test spectrum, the ability to diagnose early-stage NSCLC through LB will only increase. Besides lung cancer diagnosis, LB could potentially aid in differentiating pulmonary nodules and even indicate the anatomical site of the primary tumor, allowing for accelerated diagnosis of cancer [23,24].

## 3. Screening

Radiologic screening of high-risk adults through low-dose CT can reduce lung-cancer-related mortality by 24% in comparison to no screening [25]. Low-dose CT screening detects more lung cancers in a high-risk population in comparison to radiography (1.1% versus 0.7%, respectively) with a sensitivity of 93.8% (95% CI: 90.6–96.3) versus 73.5% (95% CI: 72.8–73.9), respectively [26]. Based on these results, lung cancer screening with low-dose CT is recommended in high-risk populations through a dedicated program [27,28,29]. However, besides the limitations of false positive results and radiation exposure in CT-graphic screening, a practical limitation of a radiographic screening program is low compliance. Those at the highest risk are less likely to participate in screening, and community-based proactive approaches are required to reach the predicted mortality reduction rates [30,31]. Only less than 5% of suitable candidates undergo such recommended screening in the US and the screening rate is increasing very slowly with time [32,33]. Therefore, there is a need for additional easily accessible screening methods for the early detection of lung cancer.

Screening for lung cancer through a blood-based test could facilitate early detection of lung cancer. ctDNA levels are very low in early-stage NSCLC; nevertheless, when using ultrasensitive ctDNA detection methods, ctDNA is present pre-treatment in most patients. Chabon et al. demonstrated that LB could serve as an initial, easily accessible screening method for lung cancer in high-risk patients. Patients with positive test results should then be referred for low-dose CT. Even though LB is at this time presumably less sensitive than low-dose CT, it has the potential to increase the total number of patients screened and reduce false positive rates, thereby potentially leading to reduced lung cancer mortality rates [17].

Other studies implemented LB to simultaneously screen for multiple (nonmetastatic) cancer types [34,35,36]. CancerSEEK reported the ability to diagnose the presence of early cancers with relatively high sensitivity (60% for lung cancer; 95% CI 55–65%) and a specificity of 99% whilst also differentiating the original organ of cancers effectively [24]. The GRAIL multi-cancer early detection study program (GRAIL LLC, Menlo Park, CA, USA) comprises of four prospective clinical trials to develop a multi-cancer early detection test and validate its performance in the real world. The first of these trials was the circulating cell-free genome atlas (CCGA) study, which was a prospective observational study to develop a plasma cfDNA-based multi-cancer detection assay [37]. This cfDNA-based blood test was able to detect multiple cancers at various stages consistently in training and test sets, as was, for example, demonstrated by the high sensitivity for the detection of lung cancer (70%). The other three clinical trials are still running, with the PATHFINDER study to be the first to return results from the multi-cancer early detection test to clinicians and the SUMMIT study to validate the test in a population of 50,000 individuals, half of whom are heavy smokers at high risk for lung cancer [38].

Further prospective studies in a large population are thus needed (and already ongoing) to confirm the clinical utility of such multi-analyte blood tests for many different cancer types at once.

## 4. Prognosis

Liquid biopsy can be used for prognostic purposes. A strong prognostic association has been described between high pre-treatment ctDNA levels (defined as a ctDNA level higher than the cohort median) and clinical outcomes in early-stage NSCLC. High pre-treatment ctDNA levels predict higher chances of recurrence and development of metastatic disease with a hazard ratio (HR) for disease recurrence of 4.48 (95% CI: 1.98–10.14; *p* = 0.0004) in stage I-III NSCLC patients and HR of 9.34 (95% CI: 2.85–30.63; *p* = 0.0004) in stage I patients [17]. High pre-treatment ctDNA levels may thus reflect the presence of sub-radiographical (micro-)metastatic disease spread.

A meta-analysis of 21 studies comprising 2143 patients showed that ctDNA detection after surgery, so-called MRD, is a strong predictor for disease relapse: HR 4.95 (95% CI: 3.06–8.02; *p* < 0.001) [39]. MRD also associates with shorter OS (HR 3.93; 95% CI: 1.96–6.77; *p* < 0.001). Inversely, longitudinally undetectable MRD can indicate a cured patient population [40]. Zhang et al. prospectively analyzed ctDNA in 261 surgically treated stage I-III NSCLC, corresponding to 652 post-surgical blood samples. Of these patients, 62.4% had stage I, 20.3% stage II, and 17.2% stage III disease. A total of 32.6% of patients had ctDNA shedding pre-operatively. The presence of pretreatment ctDNA correlates as expected with larger tumor size and higher disease stage. The investigators showed a negative predictive value (NPV) of longitudinal MRD analysis of 96.8%. In other words, 96.8% were still disease-free at the last follow-up and this predictive ability appears to be disease-stage independent [40]. The survival curve of longitudinal undetectable MRD patients, independent of disease stage, is nearly perfect and approaching that of a cured population: HR 0.02 (95% CI: 0.01–0.05; *p* < 0.001). Of note, undetectable ctDNA pre-operatively does not affect the usefulness of disease monitoring post-operatively. In this study, 41 patients with undetectable ctDNA pre-treatment showed longitudinal MRD detection. Strikingly, the few patients (n = 6) that recurred despite being MRD-negative after surgery had ctDNA detected pre-operatively.

In the phase II ABACUS trial, patients with muscle-invasive bladder cancer received two cycles of neoadjuvant PD1 inhibition (atezolizumab). This neoadjuvant regimen was able to induce a complete pathological response (cPR) in 31% (95% CI: 21–41%) of patients. These patients with cPR appeared to have more favorable clinical outcomes. ctDNA clearance was strongly correlated to pathological response and thus can be considered as a (by proxy) predictor for clinical outcome. Long-term clinical outcomes of this trial are awaited [41].

In the open label phase II LCMC3 trial, 181 patients with operable stage IB to IIIB NSCLC received two cycles of neoadjuvant atezolizumab. The primary endpoint of major pathological response (MPR; i.e., <10% viable tumor cells) was met: MPR of 20% (95% CI, 14–28%) [42]. A two-fold or larger ctDNA decrease occurred in 54% of patients after treatment with atezolizumab. ctDNA clearance was also correlated to pathological response (*p* < 0.001, r = 0.38) [43]. Preliminary results showed a trend towards improved DSF in patients showing ctDNA clearance (HR 0.25; 95% CI: 0.06–1.01 *p* = 0.035).

Forde et al. showed, in the larger (n = 358) randomized phase III CheckMate 816 study, that, in operable stage IB to IIIA NSCLC patients, neoadjuvant chemo-and immunotherapy was able to induce more pathological complete responses compared to chemotherapy alone: OR 13.94 (95% CI: 3.49–55.75; *p* < 0.001). The neoadjuvant combination therapy also resulted in longer event-free survival (EFS): HR 0.63 (95% CI: 0.43–0.91; *p* = 0.0052) [44]. pCR is a strong predictor of clinical outcome parameters such as EFS and possibly overall survival, although the data is still immature. Patients with ctDNA clearance exhibited high rates of pCR (46%). Neoadjuvant chemo- and immunotherapy had a higher rate of ctDNA clearance (56%; 95% CI, 40 to 71) than chemotherapy alone (35%; 95% CI, 21–51). Interestingly, not a single complete pathological response was noted when ctDNA clearance was absent. Event-free survival appears to be longer in patients with ctDNA clearance than in those without both in the nivolumab-plus-chemotherapy group (HR of 0.60; 95% CI, 0.20 to 1.82) as well as the chemotherapy-alone group (HR 0.63; 95% CI, 0.20 to 2.01). Long-term survival outcomes of this study are awaited as well as correlative data of ctDNA clearance and the presence of MRD on the outcome. Based on these preliminary data, a correlation between ctDNA clearance and clinical outcome can be expected.

ctDNA analysis of a smaller Chinese trial (n = 22) evaluating neoadjuvant chemotherapy, immunotherapy, or chemo- and immunotherapy showed ctDNA dynamics to be highly concordant with a pathological response (100% sensitivity). The presence of residual ctDNA pre-operatively also had a strong correlation with a higher recurrence chance after surgery [HR, 7.41; 95% confidence interval (CI): 0.91–60.22, log-rank *p* = 0.03] [45].

Therefore, pre- and post-treatment LB variables are likely to become an important tool to predict individualized patient outcomes [46].

## 5. Therapy

### 5.1. Adjuvant and Neoadjuvant Treatment in Early-Stage NSCLC

One out of four newly diagnosed lung cancers comprise (potentially) resectable disease, but, unfortunately, a significant number of patients who undergo curative surgery develop recurrence and eventually die of the (metastatic) disease [47,48]. In disease stages where adjuvant therapy is recommended, neoadjuvant chemotherapy could be used since they have similar results on overall survival [29,49]. Adjuvant and neoadjuvant chemotherapy improve 5-year overall survival by approximately 5%, compared to surgery alone. However, over 20% of these patients receiving (neo)adjuvant chemotherapy experience clinically significant acute toxicity of this therapy [50].

Several strategies have been investigated to improve outcomes in non-oncogene-driven early-stage NSCLC. Only the recent introduction of peri-operative immune checkpoint inhibition therapy (PICIT) heralds further improvement. As mentioned before, neoadjuvant nivolumab in combination with chemotherapy has been shown to improve pCR rate and EFS [44], and the addition of neoadjuvant immunotherapy did not significantly increase the incidence of adverse events. In the aforementioned CheckMate 816 study, investigators reported a grade 3 or worse treatment-related adverse event in 33.5% of patients in the neoadjuvant nivolumab-plus-chemotherapy group, compared to 36.9% of those in the chemotherapy-alone group [44].

Adjuvant immunotherapy has also been shown to significantly improve disease-free survival (DFS) [51,52]. Adjuvant atezolizumab was shown to improve DFS (0.79; 0.64–0.96; *p* = 0.020), with the treatment benefit being most pronounced in patients with (high) programmed death-ligand 1 (PD-L1) expression. Higher disease stages benefited the most from this strategy. In earlier disease stages, the improvement was less clear. Adjuvant pembrolizumab similarly reached the primary endpoint of improved DFS (HR 0.76; 95% CI 0.63–0.91; *p* = 0.0014), but no relation with PD-L1 expression was observed. Grade 3 or worse adverse events occurred relatively frequently: 11% with Atezolizumab compared to 8% in the placebo arm and 34% with Pembrolizumab compared to 26% in placebo-treated patients. However, adverse events were manageable and consistent with the known safety profile of immunotherapy.

In summary, clinical outcomes can be improved with PICIT. The impact of adjuvant immunotherapy appears less impressive than that of the neoadjuvant strategy and both come with a toxicity risk, although this is manageable but at a significant financial cost. LB might prove to be a valuable tool to better select patients who will benefit the most from PICIT.

### 5.2. Biomarker-Guided Treatment Choice

The detection of tumor mutational burden (TMB) through LB analysis will help to predict the response to immune checkpoint blockade [12,13,14,53,54]. TMB is the number of somatic variants per megabase of DNA. This biomarker predicts the response to immune checkpoint blockade. Evaluation of TMB in tumor tissue complicates the estimation of TMB due to intra-tumor heterogeneity that can be circumvented via LB in which this heterogeneity issue is absent [5]. With a growing number of clinical applications for PICIT [44,51,52], the need for validated biomarkers for response and outcome is increasing. The implications of (blood-based) TMB analysis on early-stage NSCLC management in general and PICIT in particular is, however, unclear, and many prospective trials in the metastatic settings have so far failed to position TMB as a biomarker for therapy decisions.

### 5.3. Minimal Residual Disease

As discussed above, LB could be used to evaluate the presence of MRD and possibly guide the need for adjuvant therapy. MRD refers to the presence of cancer cells that are below the limits of detection using conventional radiographic assessment after achieving complete remission but that can be detected by liquid biomarkers such as ctDNA or circulating tumor cells. The presence of postoperative ctDNA in the bloodstream predicts relapse risk [43,44,45]. Furthermore, ctDNA clearance is associated with acquiring a complete pathological response after neoadjuvant therapy. These findings suggest that MRD might be a guide as to whom should receive adjuvant therapy in the future.

In the DYNAMIC trial, patients with stage II colon cancer were randomly assigned (2:1 ratio) to have treatment decisions guided by ctDNA results or standard clinicopathological features [9]. Patients who were MRD-positive (i.e., had ctDNA detection post-operatively) were treated with adjuvant oxaliplatin-based or fluoropyrimidine chemotherapy. Patients who were MRD-negative were not treated. The primary efficacy endpoint was recurrence-free survival (at 2 years). A significantly lower percentage of patients in the ctDNA-guided group than in the standard-management group received adjuvant chemotherapy (15% vs. 28%; relative risk, 1.82; 95% CI, 1.25 to 2.65). When looking at recurrence-free survival, a ctDNA-guided strategy was not inferior to standard management (93.5% and 92.4; absolute difference of 1.1%; 95% CI, −4.1 to 6.2 (noninferiority margin, −8.5 percentage points)).

The Japanese GALAXY study confirmed the importance of MRD in surgically treated high-risk CRC [55]. The presence of MRD was the most significant prognostic factor associated with recurrence risk in patients with stage II or III CRC (HR 10.82, *p* < 0.001), but more importantly, MRD-positive patients showed the greatest benefit of adjuvant chemotherapy. Disease-free survival was significantly worse in MRD-positive CRC patients who did not receive adjuvant chemotherapy versus those that did receive chemotherapy (HR 6.59, *p* < 0.0001). Inversely, MRD-negative patients did not clearly show benefit from adjuvant chemotherapy, and similar disease-free survival was noted (HR 1.71; 95% CI 0.8–3.7, *p* = 0.16).

In the large (n = 809), randomized phase III trial IMvigor010, adjuvant atezolizumab, compared to observation in unselected muscle-invasive operable bladder cancer, did not result in a significant benefit of disease-free survival (HR = 0.89; 95% CI: 0.74–1.08); *p* = 0.2446), nor in overall survival (HR = 0.85 (95% CI: 0.66–1.09) [56]. Outcomes were analyzed in 581 patients who underwent ctDNA analysis. Of these, 34% had MRD after surgery, at the start of adjuvant therapy. Patients with MRD had improved DFS (HR 0.58; 95% CI 0.43–0.79; *p* = 0.0024) and OS (0.59 (95% confidence interval: 0.41–0.86)) in the atezolizumab arm versus the observation arm [8]. In conclusion, ctDNA-guided patient selection for adjuvant therapy in high-risk bladder cancer was prospectively proven to improve the outcome.

In NSCLC, the MERMAID trials (NCT04385368 and NCT04642469) embedded MRD in their trial design (adjuvant durvalumab in MRD-positive NSCLC patients after surgery or at the emergence of ctDNA during follow-up), but these trials were halted due to low patient accrual. Correlative data of ctDNA clearance and MRD on outcome in trials such as IMpower010, PEARLS, and ADAURA will further clarify the usefulness of ctDNA as a guide for (neo)adjuvant strategies [51,52,57].

### 5.4. Longitudinal ctDNA Monitoring

Liquid biopsy might become a tool to monitor tumor burden and for timely detection of disease recurrence [58]. A rapid decline in ctDNA levels has been described post-resection in early-stage lung cancer [59,60]. ctDNA has a potential role in detecting recurrence even before (radiologically) overt disease imaging [61]. A meta-analysis of nine studies reported a median lead time of 179 (±74) days [39]. Longitudinal monitoring after surgery or chemoradiotherapy with curative intent for stage IA-IIIB NSCLC patients (n = 88), showed a median lead time of 212.5 days between ctDNA emergence and confirmed disease progression [62]. This same study also confirmed the importance of ctDNA detection post-therapy as a predictive prognostic biomarker with a significantly higher risk of disease recurrence in case of residual ctDNA (HR 9.81; CI 95%: 4.74–20.29; *p* < 0.001).

It remains, however, unclear at present if ctDNA-guided therapeutic interventions during follow-up, e.g., initiating therapy at the time of ctDNA emergence during follow-up before radiographical relapse, will improve clinical outcome.

Another hypothetical, but promising, application of ctDNA monitoring in early-stage NSCLC is the timely identification of resistance to adjuvant chemotherapy after surgery and subsequent targeting of these resistance-emerging subclones before clinical relapse [63]. The same strategy could also be applied to patients treated with adjuvant immunotherapy or targeted agents after either surgery or chemoradiotherapy. In a future personalized strategy, only patients who are likely to recur, e.g., selected based upon the presence of MRD, are likely to receive adjuvant therapy. ctDNA monitoring during this therapy might be able to detect emerging subclones responsible for (metastatic) disease recurrence. These (sub)clones could then be targeted early during therapy, and this strategy could potentially improve the outcome or even prevent recurrence. The TRACERx study is a prospective cohort study that monitors the clonal evolution of 100 patients with early-stage NSCLC from diagnosis until death [64]. Abbosh et al. conducted a bespoke multiplex-PCR NGS approach to profile ctDNA in a subgroup of 24 patients from the TRACERx cohort. They performed blinded profiling of post-operative plasma samples to map evidence of adjuvant chemotherapy resistance, trace patients who are likely to experience recurrence of their lung cancer and track the subclonal nature of lung cancer metastases and relapse. For each patient, clonal and subclonal single-nucleotide variants (SNV) were selected to detect phylogenetic tumor branches in plasma based on their Variant Allele Frequencies (VAFs), with high sensitivity and specificity (99%) at a VAF above 0.1%. Adjuvant chemotherapy resistance seemed to be predicted by ctDNA profiling, as the number of detectable SNVs increased during adjuvant chemotherapy in patients who would later have disease recurrence, whereas SNVs diminished and eventually even disappeared in patients who remained relapse-free post-surgery and post-adjuvant-chemotherapy. In four patients who suffered NSCLC recurrence, a relapse process dominated by one specific subclone was suspected since the subclonal SNVs displayed similar VAFs to clonal SNVs from clusters confined to a single phylogenetic branch. Subclonal ctDNA analyses were validated using sequencing data acquired from metastatic tissue at the time of clinical recurrence. Analysis of relapse tissue revealed the subclonal cluster containing the SNV that gave rise to the metastatic subclone. Subclones carrying druggable oncogenic drivers can be treated with targeted agents. The use of ctDNA analysis for early detection of clinically actionable mutations after curative therapy months before radiologic relapse has also been investigated in other tumor types, e.g., stages I to III of colorectal cancer [65]. Although still in its infancy, the use of ctDNA for the study of mutational signatures to predict cancer evolution and adjuvant chemotherapy resistance and target emerging subclones might become a useful tool.

In other tumor types, such as high-risk bladder and colorectal cancer, LB has proven its importance in the early stage and treatment. In NSCLC, better patient selection through LB could be used to optimize peri-operative systemic treatment by biomarker-driven treatment selection, detection of MRD, and timely detection of recurrence.

## 6. Conclusions

Liquid biopsy is becoming a highly valuable tool to improve survival rates in early-stage NSCLC. Pretreatment, it can be implemented to enhance lung cancer screening, accelerate (early) diagnosis and further individualize prognosis in the peri-operative setting. Liquid biopsy is a predictive biomarker for response and outcome for (neo)adjuvant therapies. Moreover, LB can potentially improve patient selection for adjuvant therapy (e.g., by the detection of MRD) and aid in selecting optimal treatment regimens (e.g., by identifying therapeutical predictive biomarkers). Longitudinal MRD surveillance will enable early detection of disease relapse during follow-up, allowing for earlier directed therapeutical interventions.

Once confirmed by prospective studies, the detection of MRD post-surgery will most probably be implemented in the standard clinical care of early-stage NSCLC. The initial results of the applications of LB are promising, and, with improved sensitivity and specificity of the detection methods, the value of LB in the management of early-stage NSCLC will become mainstream.

## Figures and Tables

**Figure 1 cancers-15-02702-f001:**
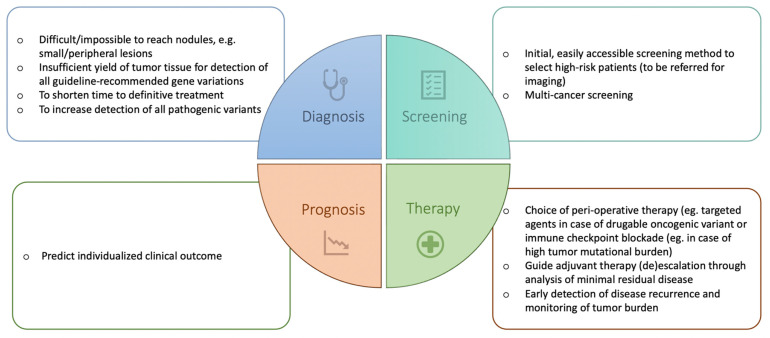
Potential current and future applications of liquid biopsy in early-stage NSCLC.

**Table 1 cancers-15-02702-t001:** Advantages and disadvantages of liquid biopsy in early-stage NSCLC.

	Advantages	Disadvantages
Diagnosis	○ Accessibility: non-invasive, good repeatability ○ Speed: shorter time to definitive treatment ○ Preservation of tumor tissue for other biomarkers ○ Increased detection of pathogenic variants	○ False negatives ○ Low sensitivity in early disease stages ○ False positives ○ Low availability ○ Cost/reimbursement
Screening	○ Accessibility: more easily accessible than screening with CT ○ No radiation exposure	
Prognosis	○ Prediction of clinical outcome ○ Presence of pre-therapy ctDNA as a proxy for disease burden ○ Dynamics of ctDNA during therapy as a proxy for therapy response and outcome ○ Presence of post-therapy ctDNA as a predictor for disease relapse	
Therapy	○ Detection of tumor mutational burden ○ Taylor adjuvant therapy presence of minimal residual disease ○ Monitoring of tumor burden/detect early relapse ○ Detection of resistant clones	

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
