# Peer review of "Liquid Biopsy in Early-Stage Lung Cancer: Current and Future Clinical Applications"

_cancers, 2023, doi:10.3390/cancers15102702_

Round 1

Reviewer 1 Report

The manuscript focuses on systemic revision of literature data about the role of liquid biopsy in early stage setting of lung cancer patients represents a timely relevant manuscript where moderate integrations should be approached to improve the readibility of this manuscript on this journal

- Introduction section seems to short to comprehensively overview all the clinical, methodological and biological aspects behind the implementation of liquid biopsy in diagnostic setting, particularly early-stage setting. I would reccomend to review this section

- Early stage setting may represent the central theme of the mansucript. Accordingly, all section should be addressed to aim this topic. I would suggest the authors to review literature data and provide more data about this topic. Pèarticularly, I would reccomend to describe GRAIL study in details focusing on outstanding data obtained from lung cancer patients.

- Liquid biopsy include a plethora of several analytes that could be managed for clinical adminsitration of soldi tumor patients. As regards, I would reccomend to investigate other promising biomarkers found in peripheral blood (e.g. CTC, miRNAs, cfDNA) in the early stage setting of lung cancer patients.

Author Response

Dear Editor and reviewer.

Thank you for the thorough reading of our manuscript. We appreciate the feedback and the opportunity to revise and resubmit. We addressed the comments raised by the reviewers and changed the manuscript accordingly.

Reply to the comments raised by Reviewer 1

Introduction section seems too short to comprehensively overview all the clinical, methodological and biological aspects behind the implementation of liquid biopsy in diagnostic setting, particularly early-stage setting. I would recommend to review this section.

We supplemented the introduction section with a description of the potential benefits of implementing liquid biopsy in early-stage NSCLC, as shown in Figure 1.

Early stage setting may represent the central theme of the manuscript. Accordingly, all section should be addressed to aim this topic. I would suggest the authors to review literature data and provide more data about this topic. Particularly, I would recommend to describe GRAIL study in details focusing on outstanding data obtained from lung cancer patients.

We supplemented the screening section with data from the GRAIL study regarding the multi-cancer early detection test, its high sensitivity for screening for lung cancer and the ongoing trials. However, the current available data on ctDNA in early stage NSCLC is mainly retrospective of nature and to our opinion all relevant clinical trials/information have been incorporated in the manuscript in our opinion. Prospective data on ctDNA in early stage RCT's are eagerly awaited (eg. results of large trials such ADAURA, Checkmate 816, IMPower010, …).

Liquid biopsy include a plethora of several analytes that could be managed for clinical adminsitration of solid tumor patients. As regards, I would recommend to investigate other promising biomarkers found in peripheral blood (e.g. CTC, miRNAs, cfDNA) in the early stage setting of lung cancer patients.

We mentioned the various biomarkers found in peripheral blood and briefly discussed their possible role in early-stage NSCLC. We would like to emphasize however that this manuscript is mainly focused on clinical use and we believe that an in depth elaboration on the different biomarker technicalities would be beyond the scope of this manuscript.

Reviewer 2 Report

This review article deals with issues related to lung cancer where liquid biopsy techniques could contribute to a significant improvement over the current situation in clinical practice.

The review is correct and comments on recent work carried out in the field of lung cancer and, in some cases, others referring to other tumors that could be extrapolated. The ideas and conclusions pointed out by the authors of the manuscript are based on these previous works and are reasonable, constituting a possible guide on where to focus the efforts to develop in the improvement and implementation of liquid biopsy techniques.

I think this review of the topic can be an good introduction for interested readers, although the subject has been object of several reviews lately (for example, PMID: 36727019, PMID: 36636413, PMID: 36596401, PMID: 36565893, PMID: 36551601, PMID: 36900221, PMID: 36797152).

As minor points to take into account, I would like to highlight some sentences from the text that I consider to be wrong and should be reviewed:

page 6, section 5.1, paragraph 1, line 1

Adjuvant and neoadjuvant treatment in early stage NSCLC”

This sentence seems to be separated from the rest of the text, as if it had been forgotten during the writing process.

page 7, section 5.1, paragraph 1, lines 8-11

The implications of (blood based) TMB analysis on early stage NSCLC management in general and in particular PICIT, is however unclear and many prospective trials in the metastatic settings have so far failed to clearly position TMB as a + biomarker for therapy decision.”

The last words of the sentence are in a different typeface and schematic format that should harmonize with the rest of the text.

Author Response

Dear Editor and reviewers,

Thank you for the thorough reading of our manuscript. We appreciate the feedback and the opportunity to revise and resubmit. We addressed the comments raised by the reviewers and changed the manuscript accordingly. 

Reply to the comments raised by Reviewer 2

As minor points to take into account, I would like to highlight some sentences from the text that I consider to be wrong and should be reviewed:

* page 6, section 5.1, paragraph 1, line 1: “Adjuvant and neoadjuvant treatment in early stage NSCLC”. This sentence seems to be separated from the rest of the text, as if it had been forgotten during the writing process.

This sentence is the first subtitle of section 5 ‘Therapy’, we have updated lay-out to make it more clear.

page 7, section 5.1, paragraph 1, lines 8-11: “The implications of (blood based) TMB-analysis on early-stage NSCLC management in general and in particular PICIT, is however unclear and many prospective trials in the metastatic settings have so far failed to position TMB as a + biomarker for therapy decision.” The last words of the sentence are in a different typeface and schematic format that should harmonize with the rest of the text.

Thank you for pointing this out to us, we have adapted it in the manuscript.

Round 2

Reviewer 1 Report

No other comments